# Using Remote Sensing Based Metrics to Quantify the Hydrological Response in a City

**Charlotte Wirion** *,† [ID], **Willy Bauwens †** and **Boud Verbeiren †** [ID]

Department of hydrology and hydraulic engineering, Vrije Universiteit Brussel (VUB), Pleinlaan 2, 1050 Brussels, Belgium
* Correspondence: Charlotte.Wirion@vub.be; Tel.: +32-2629-3036
† Current address: Pleinlaan 2, 1050 Brussels, Belgium.

**Abstract:** We propose a remote-sensing based metric approach to evaluate the hydrological response of highly urbanized areas and apply it to the city of Brussels. The model is set-up using 2 m resolution hyperspectral data. Next, it is upscaled to the city level, using multi-spectral Sentinel-2 data with 20 m resolution. We identify the total impervious area, the vegetation cover and the leaf area index as important metrics to derive a timeseries of spatially distributed net rainfall, runoff and infiltration from rainfall data. For the estimation of the actual evapotranspiration we use the potential evapotranspiration and the available water storage based on the interception, the depression storage and the infiltration. Additionally, we route the runoff to the outlet of selected sub-catchments. An important metric for the routing is the timing to the outlet which is approximated using the total impervious area and the hydrological distance to the outlet. We compare our approach to WetSpa model simulations and reach $R^2$ values of 98% for net rainfall, 95% for surface runoff, 99% for infiltration and 97% for cumulative evapotranspiration. The routing in the Watermaelbeek catchment is evaluated with discharge observations and reaches NSE values of 0.89 at a 2 m resolution and 0.88 at a 20 m resolution using an hourly timestep. At the timestep of 10 min and a 20 m resolution the NSE is reduced to 0.76. For the Roodebeek catchment we reach an NSE of 0.73 at a spatial resolution of 20 m and an hourly timestep. The results presented in this paper are optimistic for using spatial and temporal metrics retrieved from remote sensing data to quantify the water balance of urban catchments.

**Keywords:** spatial and temporal metrics; remote sensing; upscaling; hydrological response; water balance; city; urban

## 1. Introduction

Due to the highly impermeable land surface in cities, the hydrological response in an urban environment is altered: (1) reducing initial rainfall losses [1], (2) generating more surface runoff [2] and (3) altering infiltration and low flow regimes [3]. To manage the increase in frequency and volume of surface runoff, hydraulically efficient (storm)water drainage networks were created. These engineered systems alter the flow regime by leading high magnitudes of polluted stormwater to receiving water bodies  causing high flow events and limiting the infiltration and recharge to groundwater for low-flow events [3]. To improve urban water management practices, new developments focus on local measures to reduce or disconnect the impermeable surfaces and increase the urban vegetation network [4]. As a consequence, we observe a reduction in surface flow and an improvement in water quality [5]. Green infrastructure further increases the potential of retention and storage within an urban catchment [6,7]. However effects on groundwater recharge due to an increase in infiltration are more ambiguous [3,8]. Many studies exist that evaluate the local effects of new water management

practices [7,9] but in order to manage entire urban catchments or cities it is important to upscale the effects to multiple spatial scales [10,11]. To fulfil this task, a detailed description of the heterogeneous urban landcover is needed.

High resolution and hyperspectral remote sensing data combined with LiDAR data has shown its potential in characterizing urban man-made surfaces and vegetation properties [12–17]. However, hyperspectral images, airborne or from satellites, are limited to small spatial extents or medium spatial resolutions and long revisit times due to acquisition costs [18]. Therefore, an assessment of the hydrological response at the scale of a city or urban watershed still depends on multispectral satellite data. The Sentinel-2 multispectral satellite provides systematic high resolution imagery at short revisit times and therefore indicates the potential to characterize spatial variations of the urban landscape as well as temporal variations of urban vegetation [18,19]. The first research question we address in this paper is: Can the hydrological response of the urban surface be characterized using remote sensing based metrics and upscaled from field scale to a city-wide scale in order to improve water regulation services of urban ecosystems. To address this question we use airborne hyperspectral high-resolution (2 m) APEX data to characterize the urban landscape at a field-scale and then develop a spatial-metric approach to upscale the hydrological response to a coarser resolution (20 m) using Sentinel-2 data.

Spatial-metrics are commonly used to study changes and spatial relationships in urban patterns [20–23]. The use of spatial-metrics in a hydrological context is more recent and remains ambiguous [24–27]. Miller and Brewer (2018) indicate a potential to use spatial metrics in urban flood estimation on event basis and catchment-scale hydrological modelling. However, our focus in this study is not on flood estimations at a catchment's outlet but to represent the overall hydrological response (all components of the surface water balance) in the context of a city. Van Nieuwenhuyse et al., 2011 warn that the interaction between spatial metrics and the internal catchment functioning is complex and needs further understanding before such an approach can be applied to ungauged basins. Thus the second question addressed in this paper is: Which remote sensing based metrics are appropriate to upscale the hydrological response in an urban catchment? To answer this question we test different metrics with regard to their potential of (1) upscaling the hydrological response from 2 m to 20 m, (2) representing hydrological fluxes occurring on the urban surface and (3) routing the surface runoff throughout the catchment.

## 2. Materials and Methods

Within this paper we develop a generic method to use remote sensing based metrics to quantify the hydrological water balance fluxes at a city wide scale. As a city isn't corresponding to a closed catchement, we choose a micro-catchment with high resolution data coverage and discharge measurements to set-up the spatial metric approach. To validate the estimation of the different hydrological fluxes in the micro-catchment, we compare the results with the high resolution simulations of a process-based water balance model—the Water and Energy Transfer between Soil Plants and Atmosphere simulator (WetSpa; [28]). Additionally we validate the estimated surface runoff using discharge measurements at the outlet of 2 micro-catchments.

### 2.1. The Study Site

The city we focus on for our study is the Brussels capital region (see Figure 1) with an area of 715 km². In the Brussels capital region, urban areas cover 25.2% (grey & black), low vegetation such as lawn, meadows and shrubs 53.3% (bright green), high vegetation (urban trees and mixed forests) 13.5% (dark green), bare soil 7.5% (yellow) and water surfaces 0.5% (blue). The soil type in the Brussels capital region varies between sandy loam, silt loam and silt clay along the river beds. Brussels has a temperate climate with moderate temperatures and an average rainfall of 853 mm/year [29]. Thus, cloud cover is an issue in this study area.

To develop and validate the method we need closed catchments with high resolution data. We therefore develop the methodology in the Watermaelbeek (WMB) catchment, (pink) and use the

more urbanized Roodebeek (RDB) catchment (yellow) as the validation site. The APEX (Airborne Prism Experiment) coverage is illustrated in red on Figure 1. The WMB catchment has an area of 7.2 km$^2$ and is subdivided into 39 sub-catchments with an average size of 0.2 km$^2$. The elevation decreases from the southwest to the northeast and ranges from 121 m to 54 m a.s.l. The average slope is 26 degrees. For the Watermaelbeek, the predominant soil type is well-drained loamy soil. The landcover consists of 39% urban, 36% high vegetation, 17.3% low vegetation, 6.5% bare soil and 1.2% water. Within the WMB catchment operates one retention reservoir of 40.000 m$^3$.

The RDB catchment has an area of 3.3 km$^2$ and is subdivided into 18 sub-catchment with an average size of 0.2 km$^2$. The elevation decreases from the west to the east and ranges from 125 m to 42 m a.s.l. with an average slope of 32 degrees. The predominant soil type in Roodebeek is sandy loam. The landcover consists of 64% urban, 9% high vegetation, 24% low vegetation and 3% bare soil; representing a much more urbanized landcover than the Watermaelbeek. Within RDB operates a retention reservoir of 33.000 m$^3$ (representing almost double the capacity per area compared to the WMB reservoir).

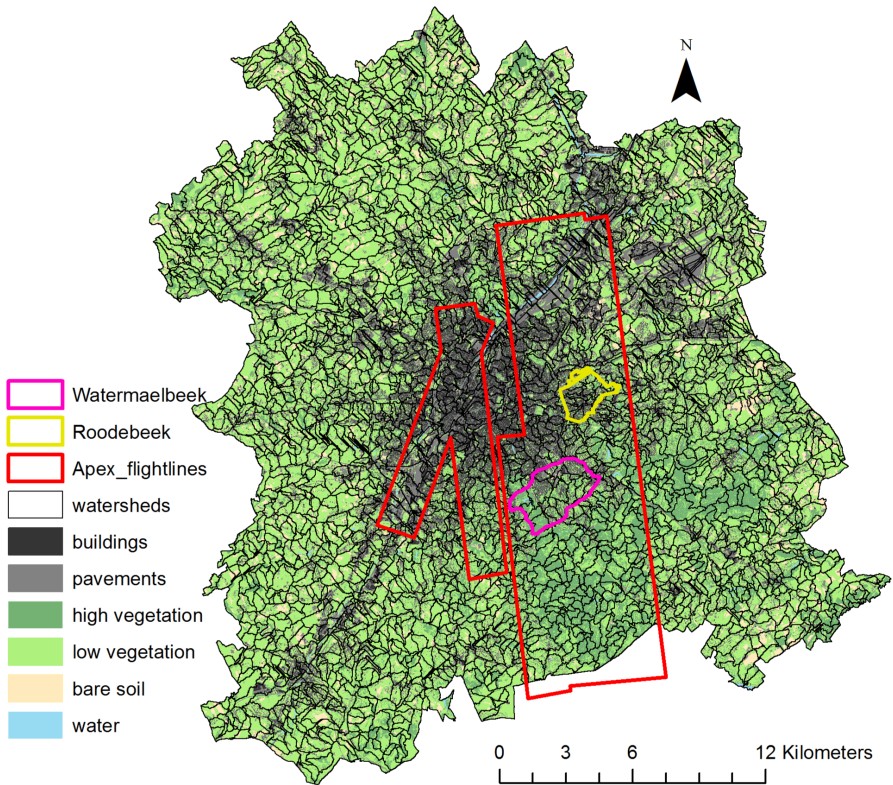

**Figure 1.** The Brussels capital region with the Watermaelbeek study site (pink), the Roodebeek validation site (yellow) and the APEX (Airborne Prism Experiment)flightlines (red).

### 2.2. The Meteorological Data

The meteorological inputs are based on hourly data for 2015 and 2016 from the Royal Meteorological Institute of Belgium (RMI) for the Uccle station in the Brussels region. The RMI measures temperature, relative humidity, precipitation, wind direction and speed, as well as global and infrared radiation data. The precipitation data are completed with pluviometer measurements within the Watermaelbeek and Roodebeek catchments using a 5 min timestep under the supervision of Flowbru, a water monitoring network for the Brussels Capital Region [29]. The potential evapotranspiration is calculated using the Penman–Monteith equation [30].

### 2.3. The Remote Sensing Data (RS)

The airborne hyperspectral APEX (Airborne Prism Experiment) image with a 2 m resolution was taken on 30 June 2015 within the framework of the Belair (2015) campaign (Belair2015 (http://belair.vgt.vito.be/content/belair-2015)). In Figure 1 the spatial coverage of the APEX image is illustrated in red. For details about the creation of the landcover map for the APEX image we refer to Priem & Canters, 2016 [13]. The Sentinel-2 satellite was launched in June 2015 and provides a multispectral dataset with 13 bands in the visible, near infrared and short-wave infrared part of the spectrum (Sentinel-2 (https://www.esa.int/Our_Activities/Observing_the_Earth/Copernicus/Sentinel-2/Introducing_Sentinel-2)) with a revisit time of 5 days. Over Brussels the spatial resolution is 20 m. We refer to Priem et al., 2017 [31] for the creation of landcover fraction maps for the Brussels capital region.

### 2.4. The Water Balance Model: WetSpa

The Water and Energy Transfer between Soil Plants and Atmosphere simulator (WetSpa) [32] allows a detailed modeling of the land surface processes. A more recent version of WetSpa, WetSpa-Python, was developed in view of the application in urban areas [33]. This version increases the flexibility of the different model components in that every physically-based process is coded in a separate module and every component can have a different spatial and temporal resolution. To account for the heterogeneous distribution of urban greenness and the seasonal effects of the vegetation, an LAI module that calculates interception storage capacity based on LAI maps has been included [14].

### 2.5. The Metrical Upscaling

The spatial coverage of the hyperspectral high resolution data is limited in space and revisit time. Therefore, we upscale the hydrological response from the micro-catchment (2 m resolution) to the city-wide scale (20 m resolution) using spatial and temporal metrics (Table 1). We consider 5 hydrological processes throughout the metric upscaling process: (1) net rainfall production, (2) surface runoff volume, (3) infiltration volume (4) cumulative evapotranspiration and (5) surface runoff routing to the outlet. With net rainfall production we refer to the rainfall reaching the ground surface after interception storage. The spatial metrics are calculated for 39 sub-catchments within the Watermaelbeek catchment at a 2 m (APEX) and a 20 m (Sentinel-2) resolution. Metrics depending on the elevation rather than the land-cover are calculated based on a 1 m resolution digital surface model (DSM) from Informatie Vlaanderen (2015) [34], which is resampled to 2 m. The temporal metrics are calculated per sub-catchment at an hourly timestep for 1 year (2015).

To select the right variables for our approach we test (a) the spatial metrics for their transferability from 2 m to 20 m resolution data and (b) the best performing configuration of metrics to represent the hydrological process in a regression. We use the ordinary least square method in a regression model. To evaluate the fit we use $R^2$ and Root Mean Square Error (RMSE).

The hydrological process is represented creating a regression equation with the chosen metrics defined at a 2 m resolution. The equation is created for processes (1)–(4) to fit WetSpa simulations whereas discharge observation data is fitted for the routing process (5). The WetSpa simulations for the Watermaelbeek catchment for 2015 at an hourly timestep and 2 m resolution are described in Reference [14]. The performance of our regression for the estimation of the discharge at the outlet is validated with the Nash-Sutcliffe efficiency (NSE) and Percent bias (Pbias) [35].

For the upscaling process, the established equations are applied using the metrics calculated based on 20 m resolution data (Sentinel-2) for the Watermaelbeek. To validate the method we check the transferability to (1) another watershed, the Roodebeek catchment, (2) another timeseries, 2016 and (3) another timestep, 10 min. An additional performance measure the ratio of the root mean square error to the standard deviation of measured data (RSR) is applied [35]. The method is then used to estimate the water balance fluxes for Brussels capital region.

**Table 1.** The metrics. The type indicates if they are distributed in space or in time. The data needed refers to the input needed to create those metrics: remote sensing (RS) data, digital surface model (DSM) retrieved from LiDAR data or meteorological datasets. The reference helps to understand the origin of the metric definition or the dataset.

| Metric | Type | Data Needed | Reference |
|---|---|---|---|
| Precipitation | Temporal | Meteorological | [29] |
| Leaf Area Index (LAI) | Temporal & spatial | RS | [36,37] |
| Vegetation cover (Veg%) | Spatial | RS | / |
| Total impervious area (TIA) | Spatial | RS | [38] |
| Directly connected impervious area (DCIA) | Spatial | RS | [38] |
| Time to outlet (tO) | Spatial | RS, DSM | / |
| Hydrological distance (hdO) | Spatial | DSM | [27] |
| Proximity index (PX) | Spatial | DSM | [24] |

## 3. Results and Discussion

### 3.1. The Regression at the Micro-Catchment Scale (2 m)

The Surface Water Balance

In a first step we compare the spatial metrics depending on RS data at 2 m and 20 m resolution for the Watermaelbeek catchment (see Figure 2). The biggest discrepancy occurs for 'directly connected impervious area' (DCIA) as at a 20 m resolution direct connections of impervious areas are lost. However for the total impervious area (TIA), vegetation cover (Veg%) and Leaf Area Index (LAI) we get a reasonable to good fit between the 2 m and 20 m resolution datasets. Therefore these spatial metrics are chosen for the regression analysis.

As a second step (b) the best performing configuration of metrics to represent the hydrological process in a linear regression are selected.

In the correlation matrix (Table 2), it is clearly shown that precipitation P highly affects net rainfall (Pnet) and Pnet is determining for surface runoff (RO) and infiltration (Infil). Based on the scatter plots between those variables (see Figure 3), we decide to use a linear regression approach for the Pnet approximation and a polynomial regression for RO and Infil. The remote sensing based metrics (LAI, Veg%, TIA) show multicollinearity. Therefore, it is decided that only one independent remote sensing based variable can be used per hydrological process. The dependencies between the hydrological processes and the remote sensing based metrics aren't so clear. Therefore, we test different configurations (sum, multiplication, division, logarithm, exponential) for the regression approaches to select the best possible fit (see Table 3).

**Table 2.** Correlation matrix (Pearson) to approximate the volume of the 3 main surface processes: net precipitation (Pnet), surface runoff (RO), infiltration (Infil). Values in bold are different from 0 with a significance level alpha = 0.05.

| Variables | Pnet | RO | Infil | P | LAI | Veg% | TIA |
|---|---|---|---|---|---|---|---|
| Pnet | **1** | | | | | | |
| RO | **0.878** | **1** | | | | | |
| Infil | **0.977** | **0.757** | **1** | | | | |
| P | **0.990** | **0.873** | **0.966** | **1** | | | |
| LAI | −0.013 | −0.077 | 0.016 | 0.023 | **1** | | |
| Veg% | −0.014 | −0.164 | 0.053 | 0.000 | **0.522** | **1** | |
| TIA | 0.014 | **0.165** | −0.054 | 0.000 | **−0.497** | **−0.982** | **1** |

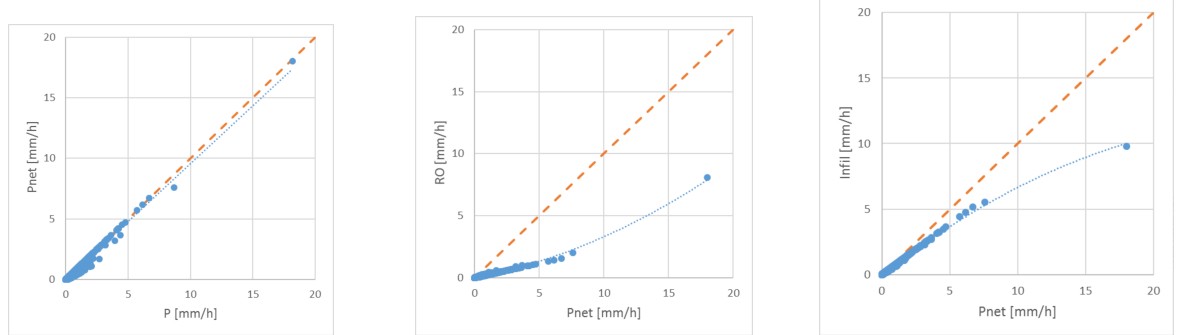

**Figure 2.** Spatial metrics for the 39 sub-catchments of the Watermaelbeek. Differences at a 2 m versus 20 m resolution for directly connected impervious area (DCIA), total impervious area (TIA), Vegetation cover (Veg%) and maximum leaf area index (LAI max).

**Figure 3.** Scatterplots for main correlated variables (P, Pnet, RO, Infil) simulated with WetSpa as averages over the whole WMB catchment. P and Pnet show a linear correlation whereas Pnet shows a polynomial correlation with RO and Infil.

**Table 3.** Linear regression for Pnet, polynomial regression of RO and Infil , sample number *n* = 46,059, N being P for the calculation of Pnet respective Pnet for the runoff and infiltration process. The best performance measures are in bold.

| | $R^2$ [%] | | | RMSE [mm/h] | | |
|---|---|---|---|---|---|---|
| | **Pnet** | **RO** | **Infil** | **Pnet** | **RO** | **Infil** |
| N | 98.1 | 83.4 | 96.8 | 0.131 | 0.187 | 0.264 |
| N + LAI | **98.2** | 84.1 | 96.9 | 0.127 | 0.185 | 0.265 |
| N + Veg% | 98.1 | 86.1 | 97.3 | 0.131 | 0.181 | 0.268 |
| N + TIA | 98.1 | 86.1 | 97.3 | 0.131 | 0.181 | 0.268 |
| N × LAI | 64.0 | 88.1 | 97.7 | 0.565 | 0.245 | 0.240 |
| N × Veg% | 83.0 | 94.3 | 98.9 | 0.389 | 0.385 | **0.188** |
| N × TIA | 72.2 | **94.7** | **99.0** | 0.496 | **0.073** | 0.335 |
| N /LAI | 88.1 | 87.6 | 97.6 | 0.325 | 0.117 | 0.339 |
| N /Veg% | 82.5 | 91.1 | 98.3 | 0.394 | 0.095 | 0.338 |
| N /TIA | 12.5 | 87.2 | 97.5 | 0.881 | 0.198 | 0.260 |
| N + LOG (LAI) | **98.2** | 77.5 | 96.9 | **0.126** | 0.185 | 0.265 |
| N + LOG (Veg%) | 98.1 | 85.7 | 97.3 | 0.131 | 0.182 | 0.267 |
| N + LOG (TIA) | 98.1 | 85.6 | 97.2 | 0.131 | 0.182 | 0.267 |
| N + EXP (LAI) | 98.1 | 83.8 | 96.9 | 0.129 | 0.186 | 0.264 |
| N + EXP (Veg%) | 98.1 | 86.2 | 97.3 | 0.131 | 0.181 | 0.268 |
| N + EXP (TIA) | 98.1 | 86.0 | 97.3 | 0.131 | 0.181 | 0.268 |

Based on the results presented in Table 3, we create the regression equations suited for each hydrological process:

$$\text{Pnet} = 0.98 \times P - 0.1 \times log(\text{LAI}) (n = 46{,}059, R^2 = 98.2\%, RMSE = 0.137\,\text{mm/h}) \quad (1)$$

The regression equation for net rainfall results in a correlation between the WetSpa simulation and the regression of 98.2% and a RMSE of 0.137 mm/h for 46,059 data points. The regression is calculated based on the LAI and the precipitation per time step and sub-catchment whenever the precipitation is not 0 mm. However, as Pnet is the most important step where inaccuracy induces errors in the surface runoff and infiltration estimation, we decide to improve the results by subdividing the equations into leafless and leafy as well as below and above averaged rainfall amounts :

Leafless: LAI < 1

$$\text{Pnet} = 0.04 + 0.99 \times P - 0.09 \times \text{LAI}(n = 44{,}444, R^2 = 99.2\%, RMSE = -0.092\,\text{mm/h}) \quad (2)$$

Leafy: LAI > 1

Below mean rainfall: $P < 0.6$ mm/h

$$\text{Pnet} = 0.03 + 0.72 \times P - 0.05 \times log(\text{LAI})(n = 1103, R^2 = 60.2\%, RMSE = 0.0931\,\text{mm/h}) \quad (3)$$

Above mean rainfall: $P > 0.6$ mm/h

$$\text{Pnet} = -0.02 + 0.98 \times P - 0.09 \times \text{LAI}(n = 506, R^2 = 96.7\%, RMSE = 0.321\,\text{mm/h}) \quad (4)$$

By implementing this subdivision you can see on Figure 4 that all watersheds get closer to the 1:1 line. The $R^2$ value is 98.3% and RMSE 0.106 mm/h. We therefore use the composite of 3 equations to define Pnet as input for the surface runoff and infiltration processes.

For the runoff equation we select the multiplication of Pnet and TIA for the polynomial regression equation:

$$\text{RO} = \text{Pnet} \times (0.01 \times \text{Pnet} + 0.47 \times \text{TIA} + 0.02)(n = 46{,}059, R^2 = 94.7\%, RMSE = 0.07\,\text{mm/h}) \quad (5)$$

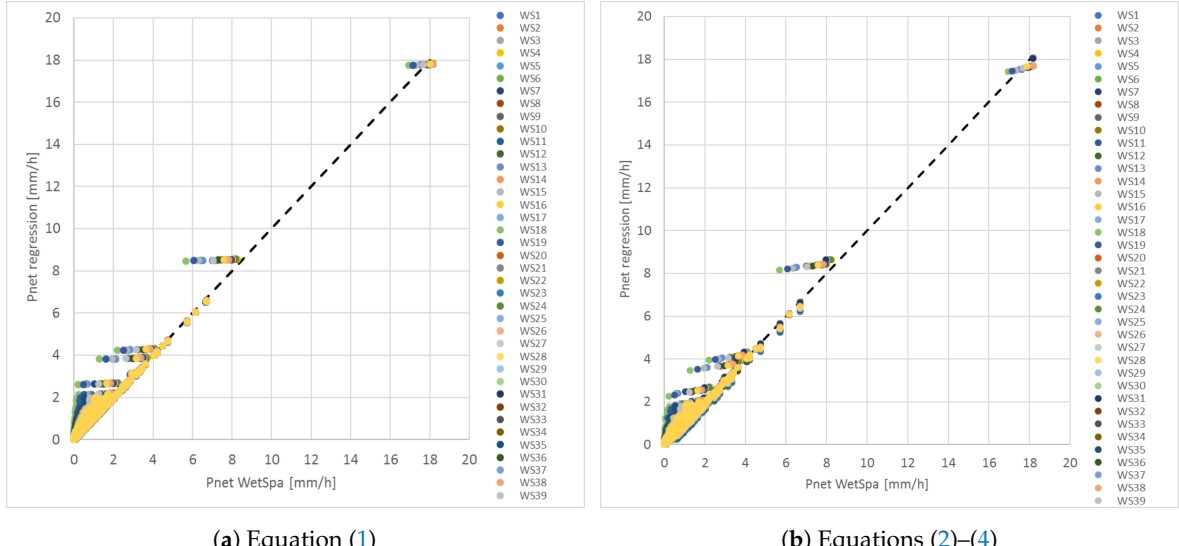

(**a**) Equation (1)        (**b**) Equations (2)–(4)

**Figure 4.** Pnet regression: (**a**) in one step (**b**) with subdivision for the 39 sub-catchments (WS) of the Watermaelbeek.

The regression equation for surface runoff (RO) resulted in a correlation between the WetSpa simulation and the regression of 94.7% and a RMSE of 0.07 mm/h for 46,059 data points (see Figure 5). The regression was calculated based on a polynomial function of the net precipitation per time step and sub-catchment and TIA per sub-catchment whenever the precipitation is not 0 mm. The results show that all sub-catchments perform similarly well. The discrepancies increase for the highest rainfall event in 2015 (18.18 mm/h) but no clear trend or bias can be identified. As a comparison, we also applied the simplified version of the Wallingford urban sub-catchment model [39] using TIA to our dataset and reach an $R^2 = 0.63$ and a RMSE of 0.10 mm/h.

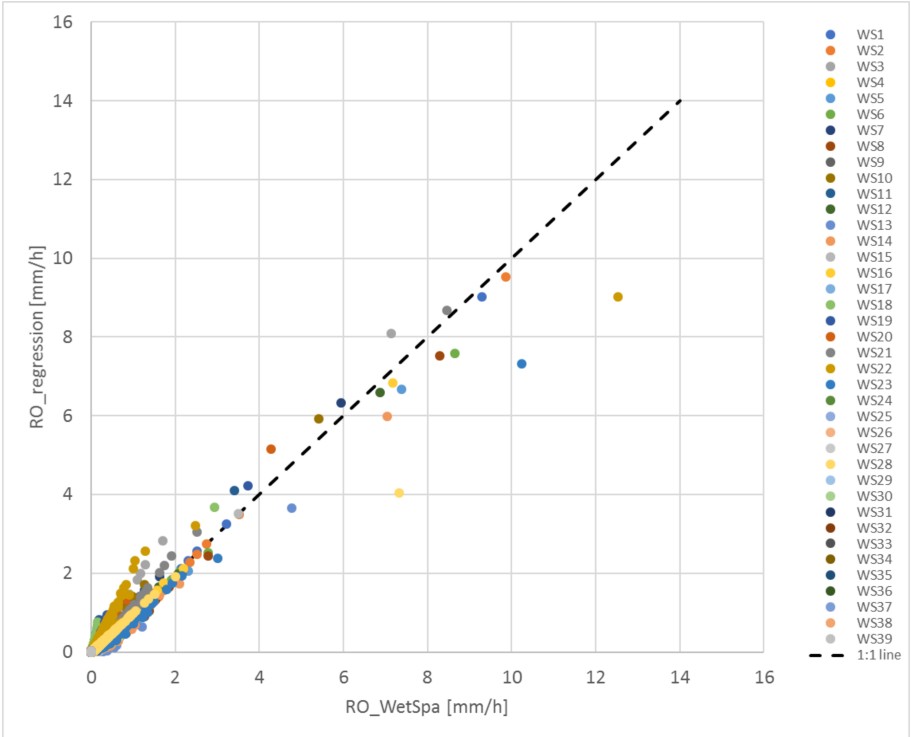

**Figure 5.** Surface runoff regression for the 39 sub-catchments (WS) of the Watermaelbeek.

For the infiltration process we use the multiplication of Pnet and Veg% as this combination reaches the lowest RMSE (with little changes in $R^2$):

$$\text{Infil} = \text{Pnet} \times (0.55 - 0.01 \times \text{Pnet} + 0.49 \times \text{Veg\%})(n = 46{,}059, R^2 = 98.9\%, RMSE = 0.08 \, \text{mm/h}) \quad (6)$$

The regression equation for infiltration results in a correlation between the simulation and the regression of 98.9% and a RMSE of 0.08 mm/h for 46,059 data points. The regression is calculated based on a polynomial function of the precipitation per time step and sub-catchment and Veg% per sub-catchment whenever the precipitation is not 0 mm. Similar to surface runoff we see on Figure 6 a bigger deviation from the 1:1 line for the biggest rainfall event (18.18 mm/h). We found that the watersheds deviating mostly from the 1:1 line and predicting higher infiltrations with the regression than with the simulations are all situated on silty soils with higher field capacities (WS 22, 23, 24, 25, 26, 27). However, adding the field capacity parameter to the regression doesn't improve the results of the equation (Infil = Pnet $\times$ (0.12 $\times$ fieldcap $-$ 0.01 $\times$ Pnet + 0.49 $\times$ Veg%) ($n = 46{,}059$, $R^2 = 98.9\%$, RMSE = 0.08 mm/h)).

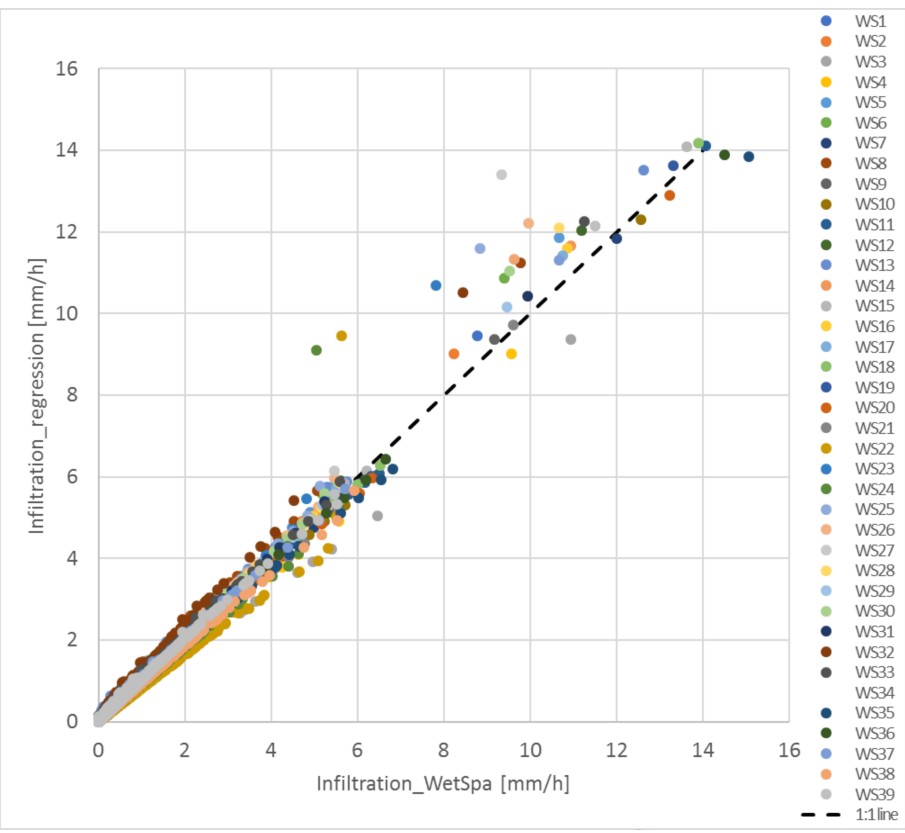

**Figure 6.** Infiltration regression for the 39 sub-catchments (WS) of the Watermaelbeek.

The water balance on the urban surface can be expressed as: Precipitation = surface runoff + infiltration + interception storage + depression storage. The water stored on the surfaces via interception and depression storage and in the soil via infiltration is available for evapotranspiration. To estimate the performance of our model we compare the measured precipitation with the surface runoff and infiltration from the regression approach (see Figure 7a). We reach an $R^2$ value of 0.99 and a RMSE of 0.05 mm/h. The underestimation of rainfall is due to the initial water storage via interception and depression storage reducing net rainfall before surface runoff and infiltration occurs. We tested several parameters to directly approximate interception storage (antecedent rainfall index, rainfall intensity, wind speed, potential evapotranspiration, LAI) but we were not able to create a regression equation with acceptable $R^2$ and RMSE values. We therefore propose to estimate interception and

depression storage with the difference between measured rainfall and regressed surface runoff and infiltration. If we compare this difference to the simulated interception and depression storage from the WetSpa model we get a good fit ($R^2$ = 97.8%, RMSE = $7.6 \times 10^{-3}$ mm/h) (see Figure 7b). The biggest differences occur for low interception values as the storage highly depends on if we are at the beginning or the end of an event. This is, for the reasons explained above, not differentiated with the regression approach.

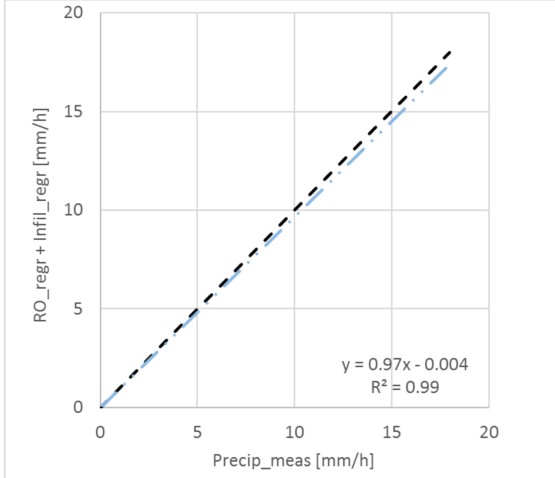
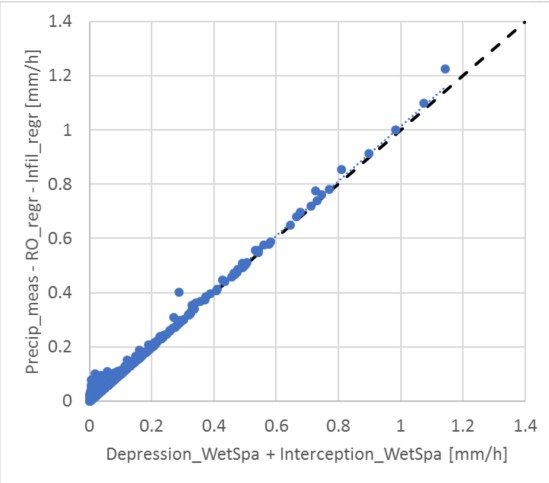

**Figure 7.** The water balance for the Watermaelbeek (WMB) catchment. (**a**) Water balance without interception and depression storage (precipitation vs. the regressed surface runoff and infiltration averaged over the WMB catchment) and (**b**) validation of estimated interception and depression storage from the regression (precipitation minus the regressed surface runoff and infiltration) with the WetSpa simulations.

### 3.2. The Evapotranspiration (ET)

ET is important in an urban context for its cooling effect on the urban climate ([40,41]). Estimating ET in an urban context remains challenging as its dependencies on rainfall, weather data (potential ET, PET) and urban vegetation can vary between cities according to the distribution of urban fabrics and micro-climates [42]. In our study, we tested the dependencies of ET on rainfall, PET, wind, TIA, VEG% and LAI. The highest dependencies were found for PET ($r$ = 0.68) and maximum LAI ($r$ = 0.69). This is partly confirmed by Smithers et al., 2018 [43], who highlight the importance of leaf area in respect to urban cooling.

In Figure 8, we illustrate how ET (green) is limited by PET (blue) until the vegetation season starts (illustrated by LAI (red)). During the vegetation season, the limitations for ET are coming from water availability/storage and vegetation. We therefore calculate storage based on the regressed infiltration, interception and depression which are dependent on P, Veg% and LAI.

$$Storage_{t+1} = \text{Infil}_t + Interc_t + Depr_t = \text{Infil}_t + (P - \text{Pnet})_t \tag{7}$$

However, with Equation (7) we don't estimate the duration of storage through dynamic filling (by rainfall) and emptying (by ET) as this deviates from the aim of this publication. Therefore, the regression equation for ET is established using the cumulative ET over the year 2015 at an hourly timestep (t) by combining storage and PET:

$$\text{ET}_t = 16 \times Storage_t \times \text{PET}_t - 1.12 \, (n = 341{,}562, R^2 = 0.97, RMSE = 6.67\,\text{mm/h}) \tag{8}$$

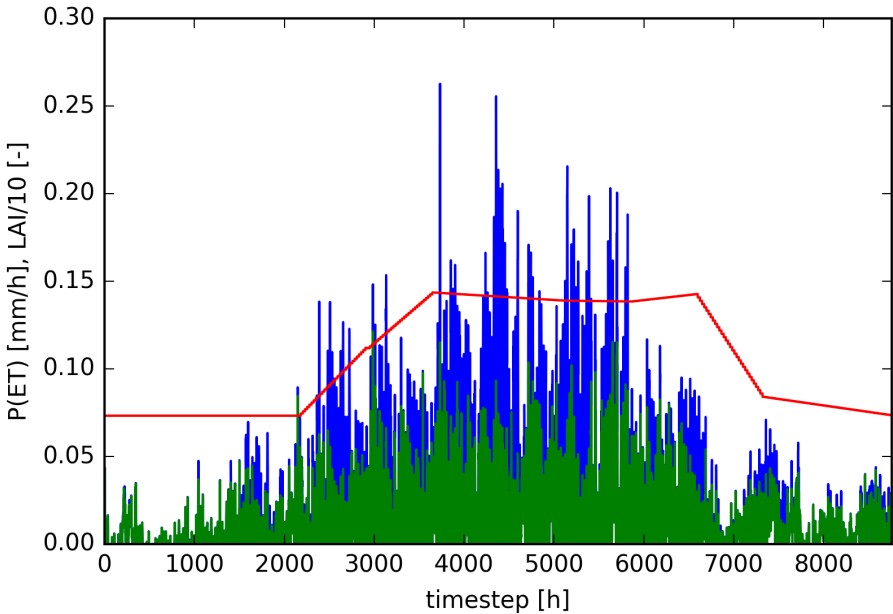

**Figure 8.** ET (green), PET(blue) and LAI/10 (red) for the Watermaelbeek in 2015 (hourly timestep).

The good fit illustrated on Figure 9 confirms that the regression method is able to reproduce the cumulative ET similarly well to the dynamic simulations with WetSpa.

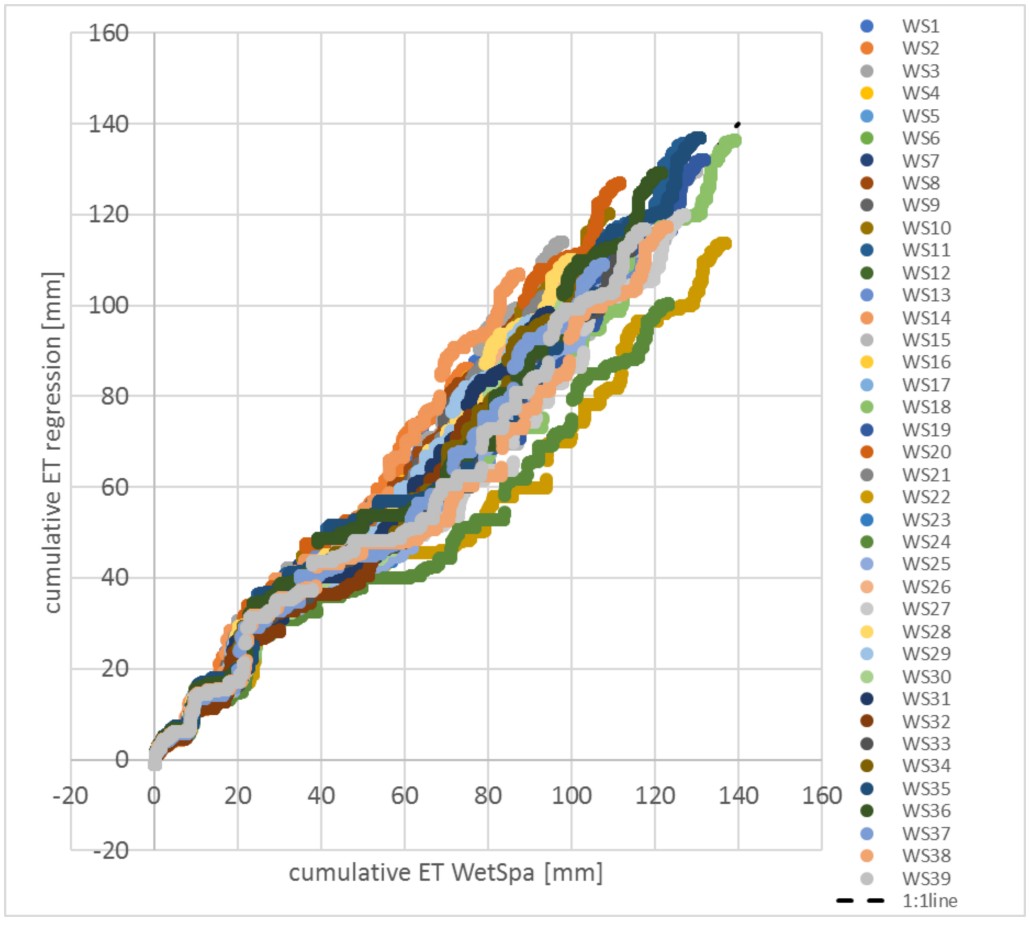

**Figure 9.** Cumulative ET: WetSpa simulation vs. regression for the 39 sub-catchments (WS) of the Watermaelbeek.

### 3.3. The Routing

To route the runoff volume to the outlet we try to approximate the timing to the outlet tO. Within the WetSpa model this timing is calculated in a physically-based manner using the flow direction and flow velocity. The flow direction is linked to the DEM whereas the flow velocity is deduced from the Manning roughness coefficient. To define which metrics can be used for the routing process we investigate the correlation between the flow time to the outlet (tO), the Manning roughness, the total impervious area (TIA), the vegetation cover (Veg%), the hydrological distance to the outlet (hdO) and the proximity index (PX). The correlation matrix (Table 4) indicates an important correlation between tO and hdO (85%). The correlation between the surface roughness and TIA and Veg% is also significant, whereas the correlation with PX is limited. The timing to outlet (tO) is thus calculated with a linear regression method using hdO and TIA as metrics (see Figure 10):

$$tO = 0.3 - 0.25 \times TIA + 0.38 \times hdO(n = 39, R^2 = 0.78, RMSE = 0.08\,\text{h}) \tag{9}$$

**Table 4.** Correlation matrix (Pearson) for the routing process. Values in bold are different from 0 with a significance level alpha = 0.05.

| Variables | tO | Manning Roughness | TIA | Veg% | hdO | PX |
|---|---|---|---|---|---|---|
| tO | **1** | | | | | |
| Manning roughness | **0.686** | **1** | | | | |
| TIA | **−0.685** | **−0.950** | **1** | | | |
| Veg% | **0.677** | **0.975** | **−0.982** | **1** | | |
| hdO | **0.849** | **0.598** | **−0.570** | **0.571** | **1** | |
| PX | −0.110 | **−0.428** | **0.440** | **−0.467** | −0.014 | **1** |

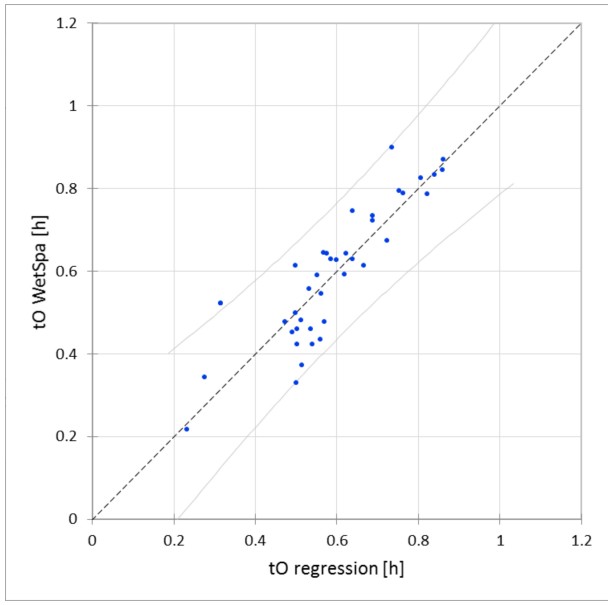

**Figure 10.** Timing to outlet : regression method vs. WetSpa simulations.

The routing process is translated into 3 equations where we (1) first evaluate the timing needed to reach the outlet (tO). If tO > 1 than the runoff accumulates over tO timesteps before reaching the outlet. Equation (10) is inspired by the Antecedent Precipitation Index [44] and indicates that the runoff at timestep t counts less than the accumulation of runoff from earlier timesteps (t -> tO). Then (2) we assume that further away sub-catchments (hdO > 1) always need longer time to the outlet than closer by catchments (Equation 12). Finally (3) the runoff volume of sub-catchments with tO < 1 and hdO < 1 reach the outlet at the same timestep t (Equation 13). The routing is expressed with the following equations:

(1) More than 1 timestep needed for runoff to reach the outlet (tO > 1):

　　-> Accumulation of runoff (*ARO*) over [tO] timesteps:

$$ARO_{t->tO} = ARO_{t->tO} + (RO_{t->tO} + (\frac{[tO_t]^1}{tO_t} + \ldots + \frac{[tO_t]^{tO}}{tO_t}) \times RO_t) \times tO_{mean} \tag{10}$$

$$Q_t = ARO_t \times PX \times k \tag{11}$$

(2) Less than 1 timestep needed for runoff to flow to the outlet (tO < 1):

　　-> Sub-catchment further away (hdO > 1) need more time (tO) to flow to outlet:

$$Q_t = (RO_{t-1} \times tO + RO_t \times (1 - tO)) \times k \times PX \tag{12}$$

(3) Less than 1 timestep needed (tO <1) and sub-catchment is close to outlet (hdO <1):

　　->Runoff reaches outlet at the same timestep:

$$Q_t = RO_t \times k \times PX \tag{13}$$

where:

*Q*: discharge at the outlet [m$^3$/s]
*ARO*: accumulation of runoff [m/h]
RO: runoff volume (from regression analysis) [m/h]
*t*: timestep [h]
tO: timing to outlet [h]
tO$_{mean}$ : mean timing to outlet for study area [h]
[tO]: integer value of the timing to outlet [h]
PX: proximity index [m$^2$]
*k*: scaling parameter for watershed sizes [-]

　　Due to potential differences in sub-catchment sizes a scaling parameter *k* is included in the equation to calibrate the order of magnitude of the discharge estimation. For the Watermaelbeek at an hourly timestep, *k* = 0.03. As the proximity index (PX) is identified as an important metric to approximate runoff at the outlet [24,27], we use PX instead of the sub-catchment area in our equations; however it is good to mention that similar results are reached using the sub-catchment area instead of PX. Further, for the comparison of regressed discharge with observed discharge, we filter out the dry-weather-flow (on average 0.09 m$^3$/s) from the observed discharge. On Figures 11 and 12 the observed discharge (yellow) is compared to the regressed discharge (grey). Figure 11 illustrates the behaviour of the regression method for common events throughout 2015, where we see an underestimation of the discharge to the outlet leading to an overall Pbias of 23% which is within the acceptable level (±25%, according to Reference [35]). The general underestimation of events occurs because we balance out the effect of reservoir occurrences for peak events. In Figure 12a we illustrate the effect of the reservoir for bigger rainfall events which we cannot capture with our regression method. The reservoir diminishes the imminent peak discharge and then releases the water continuously over a longer time period. Due to the difficulties to account for the reservoir occurrences with our regression method we reach an NSE of 0.58. Therefore, we modified the discharge timeseries for 16 identified reservoir occurrences within 2015 by adding the released discharge after reservoir occurrences to the peak (see Figure 12b). After accounting for the reservoir within the observed discharge timeseries we reach an NSE of 0.89 and reduce the Pbias to 12%.

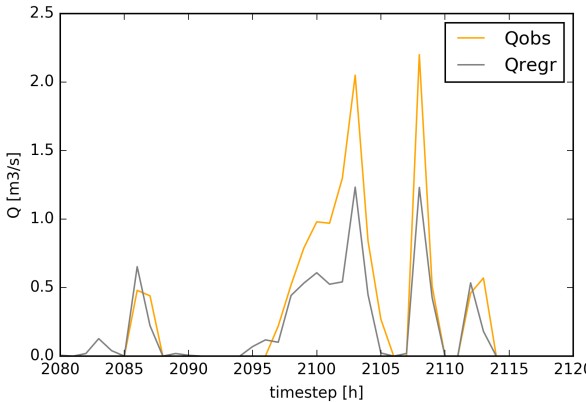

**Figure 11.** Discharge at outlet for rainfall events in March 2015. Observed discharge in yellow and regressed discharge in grey.

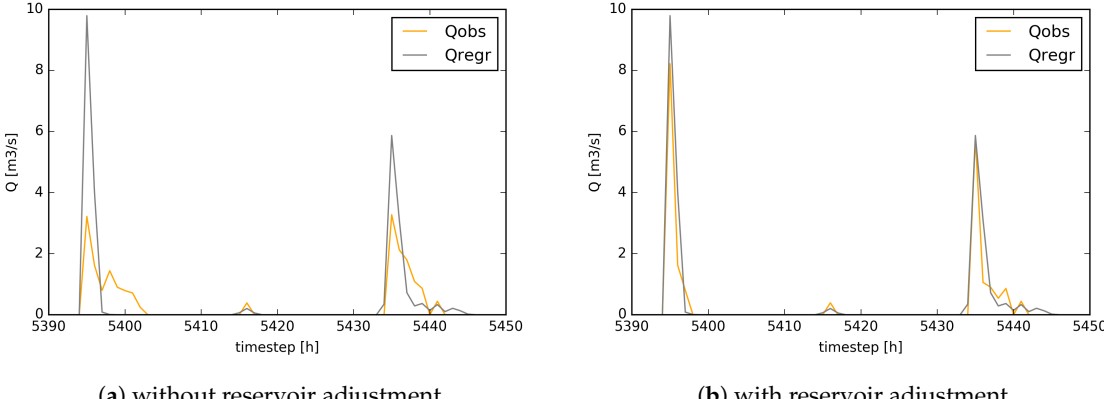

(**a**) without reservoir adjustment

(**b**) with reservoir adjustment

**Figure 12.** Observed (yellow) and regressed (grey) discharge at the outlet for bigger rainfall events in August 2015 with (**a**) reservoir occurrences and (**b**) adjusted reservoir occurrences

### 3.4. The Spatial Validation (20 m Resolution)

To upscale the metric approach from a field scale with high resolution data to a wider scale with lower resolution data, we use the Sentinel-2 data at 20 m resolution to define spatial and temporal metrics (TIA, VEG, LAI). We then use the same calculation equations as above to estimate net rainfall, surface runoff, infiltration, evapotranspiration and discharge at the outlet. For the Watermaelbeek, the correlations for the water balance components between 2 m and 20 m resolutions are very high (see Figure 13). At a 20 m resolution, and using the same k value as at 2 m resolution, the regressed discharge fits the observed discharge with NSE = 0.78 and a Pbias = 6.7%. The difference between the 2 m resolution discharge and the 20 m resolution discharge is minor ($R^2$= 99.98%, NSE = 0.997, Pbias = −6%) indicating slightly higher discharge values at a 20 m resolution. Those results make us confident in using Sentinel-2 images combined with the regression method to estimate urban surface fluxes and discharge in a city.

To further test the approach we apply the method to a different catchment in Brussels: the Roodebeek which is more urbanized and has a steeper average slope. For the Roodebeek catchment, we also filter out dry weather flow and use a factor $k$ = 0.015 due to the smaller size of the Roodebeek catchment.Further we adjust 3 reservoir occurrences using the same approach as with the Watermaelbeek catchment (4 May 2015, 5 June 2015, 13 August 2015). However, as the relative capacity of the RDB reservoir is higher than the WMB reservoir we expect a bigger influence on the discharge for the Roodebeek catchment. The visual fit (Figure 14) and the measures of fit are good (NSE = 0.73, Pbias = 11.5%, RMSE= 0.08 m$^3$/s, $R^2$ = 0.76)

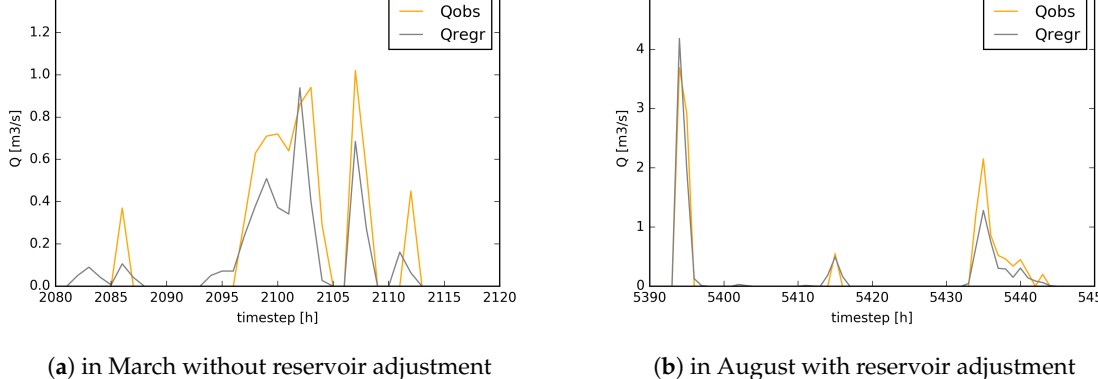

(**a**) net rainfall: $R^2$ = 99.9%, RMSE = 0.01 mm/h　　　(**b**) surface runoff: $R^2$ = 99.9%, RMSE = 0.007 mm/h

(**c**) infiltration: $R^2$ = 98.6%, RMSE = 0.036 mm/h　　(**d**) evapotranspiration: $R^2$ = 98.8%, RMSE = 4.45 mm/h

**Figure 13.** Hydrological surface fluxes: 2 m vs. 20 m regression method.

(**a**) in March without reservoir adjustment　　　　　(**b**) in August with reservoir adjustment

**Figure 14.** Discharge at the outlet, observed (yellow) vs. regressed (grey) for the Roodebeek catchment (20 m, 1 h) in (**a**) March and (**b**) August 2015.

*3.5. Validation for Different Temporal Resolutions*

Additional to the spatial validation with a different watershed we also want to validate our method at a temporal scale. To do so we first apply the method to the 2016 timeseries, next we downscale the timestep to 10 min which we consider more appropriate in an urban runoff context. We keep the same *k* value (*k* = 0.03) for the Watermaelbeek catchment. In Table 5 the performance measures of both temporal validation scenarios are presented. Transposing the spatial metric approach to the 2016 timeseries slightly decreases the goodness of fit. One explanation therefore is the increased number of reservoir occurrences in 2016 (34 in 2016 vs. 16 in 2015) due to higher precipitation in 2016 (942.3 mm in 190 days in 2016 vs. 736.7 mm in 198 days in 2015). Due to the difficulty in identifying the release of the water from the reservoir, we were only able to account for the major reservoir occurrences on 7 January 2016 and 7 June 2016. The Pbias indicates a higher overestimation in 2016 compared to 2015 (e.g., the biggest observed peak is 9.93 m$^3$/s whereas the regressed peak reaches 13.75 m$^3$/s). The RMSE and RSR performance measure indicate that the error is higher in 2016 than 2015 but lower than the standard deviation of the measurements (std = 0.23 m$^3$/s). At a 10 min timestep, the performance is lower than at an hourly timestep but still good. We have to note that at a 10 min timestep we only accounted for the reservoir for the main peak event in August 2015 (Figure 15a) as the dataset is too big to consistently account for reservoirs. The Pbias confirms an overestimation of peak events seen on Figure 15. Further the RMSE and RSR performance measure indicate that the error is higher for the 10 min compared to the hourly timestep but still remains slightly lower than the standard deviation of the measurements (std = 0.23 m$^3$/s). On Figure 15, it is interesting to see that the timing of the flow to the outlet corresponds to the measurements which confirms the routing in our approach.

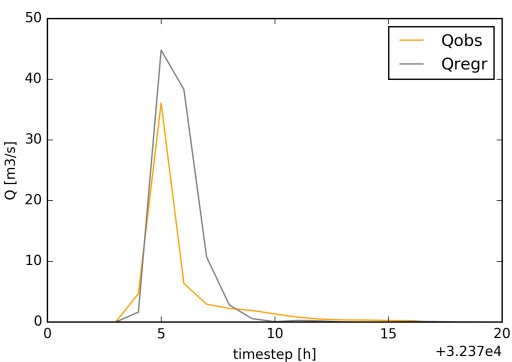
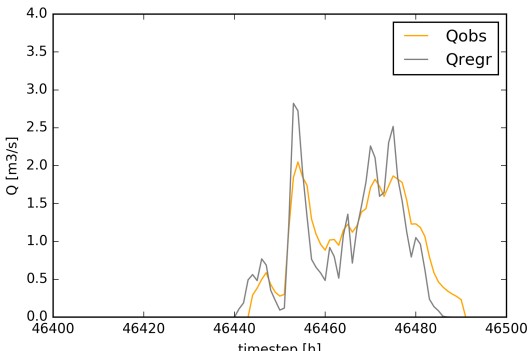

(**a**) in August with reservoir adjustment                      (**b**) in November without reservoir adjustment

**Figure 15.** Discharge at the outlet, observed (yellow) vs. regressed (grey) for the Watermaelbeek catchment (20 m, 10 min) in (**a**) August and (**b**) November 2015.

**Table 5.** Statistical performance measures to evaluate the discharge estimation after correcting for the above-mentioned reservoir occurrences [35].

| Statistical Measure: | NSE [-] | Pbias [%] | RMSE [m$^3$/s] | RSR [-] |
|---|---|---|---|---|
| Good performance | >0.5 | <±25% | / | <1 |
| WMB 2 m, 1 h (2015) | 0.89 | 12 | 0.09 | 0.45 |
| WMB 20 m, 1 h (2015) | 0.88 | 6.7 | 0.09 | 0.45 |
| WMB 20 m, 1 h (2016) | 0.80 | 17 | 0.15 | 0.67 |
| WMB 20 m, 10 min (2015) | 0.76 | 11.1 | 0.205 | 0.89 |

*3.6. Upscaling to a City-Wide Scale*

Within this paper we aim at quantifying the hydrological response of a city using spatial and temporal metrics retrieved from remote sensing data. We validated the approach with observed

discharge data from 2 closed catchments, the Watermaelbeek and the Roodebeek catchment, by routing the estimated runoff volume to the outlet using spatial metrics. We don't aim at simulating pipe flow routing for an entire city (not corresponding to a closed catchment) but limit the method to estimating the water balance components at a city wide scale. The proposed approach quickly quantifies the storage-, evaporation- and surface runoff volume distributed over a city or an area of interest within the city at different temporal scales. Figure 16 illustrates the water balance of Brussels using an hourly timestep for an event in summer and winter. 24 mm of rainfall within 8 h are measured during the summer event and 23 mm of rainfall within 41 h are measured during the winter event. Due to the higher rainfall intensity in summer we reach a higher net rainfall (Figure 16a: 98% vs. Figure 16b: 95%). However as the LAI is higher for forested areas in summer the Sonian forest in the South-East of Brussels shows lower net rainfall during this period (Figure 16a: 83% vs. Figure 16b: 91%). Further, the summer event produces more runoff over a larger area (Figure 16c,d whereas more infiltration occurs during the winter event (Figure 16f). In winter we reach very low ET values (<2 mm) over Brussels whereas in summer ET reaches >10 mm in the Sonian forest (Figure 16g,h).

Figure 16 illustrates as well the potential and limitations of this approach. It is suited to evaluate the effect of land use or climate change on the water balance in a city or neighborhood. Land use change is accounted by the remote sensing based metrics and climate change effects the water balance if the meteorological input is adapted. In other words, the proposed method can be used to quantify the water balance for different urban design scenarios to evaluate their effect not only on, for example, flood mitigation via reducing surface runoff but also, for example, on the cooling effect of pervious pavements and urban vegetation through evapo(transpi)ration [45].

We therefore hope that the proposed method is applied to the planning of sustainable future cities. As the metrics are mainly based on freely available high resolution remote sensing images, we propose to apply this method to other cities worldwide. To do so, it is important to first validate the method using closed catchments within the area with available observation data and/or a validated water balance simulation. Additionally to the free remote sensing products, you also need the meteorological input for your area of interest as well as a digital elevation model for the routing part of the method.

Compared to data-driven or physically based models, we position our method between the 2 approaches. Data-driven models are useful tools for the real-time forecasting of, for example, flooding, as the computational time is close to 0. However, scientists criticize the loss of a 'physical meaning' as the data driven model does not necessarily account for preserving a correct mass balance [46]. On the other side, process- or physically based models aim at gaining a better physical understanding of hydrological fluxes. However, the parameterization of those models is limited to the quantity and quality of the available data [46]. Another drawback is the computational efficiency when applying the modelling tools to heterogeneous urban catchments using a high spatial and temporal resolution [47]. The remote sensing based metrics used here to characterize the urban landscape add a 'physical meaning' to a very computational efficient approach to quantify the urban water balance.

We use optical and LIDAR remote sensing sources to create the metrics used in the regression. However, we see potential in adding microwave remote sensing in order to better quantify interception and depression storage, a short-coming of our approach. Microwave remote sensing has largely been used to estimate soil moisture showing draw-backs such as low spatial resolution and revisit frequency and sensitivity to soil roughness and vegetation [48]. Especially in urban areas the backscatter contamination in microwave images reduces the potential for retrieving surface moisture [49]. However, new satellites with higher spatial resolution and revisit frequency indicate a potential to retrieve moisture of urban land surfaces.

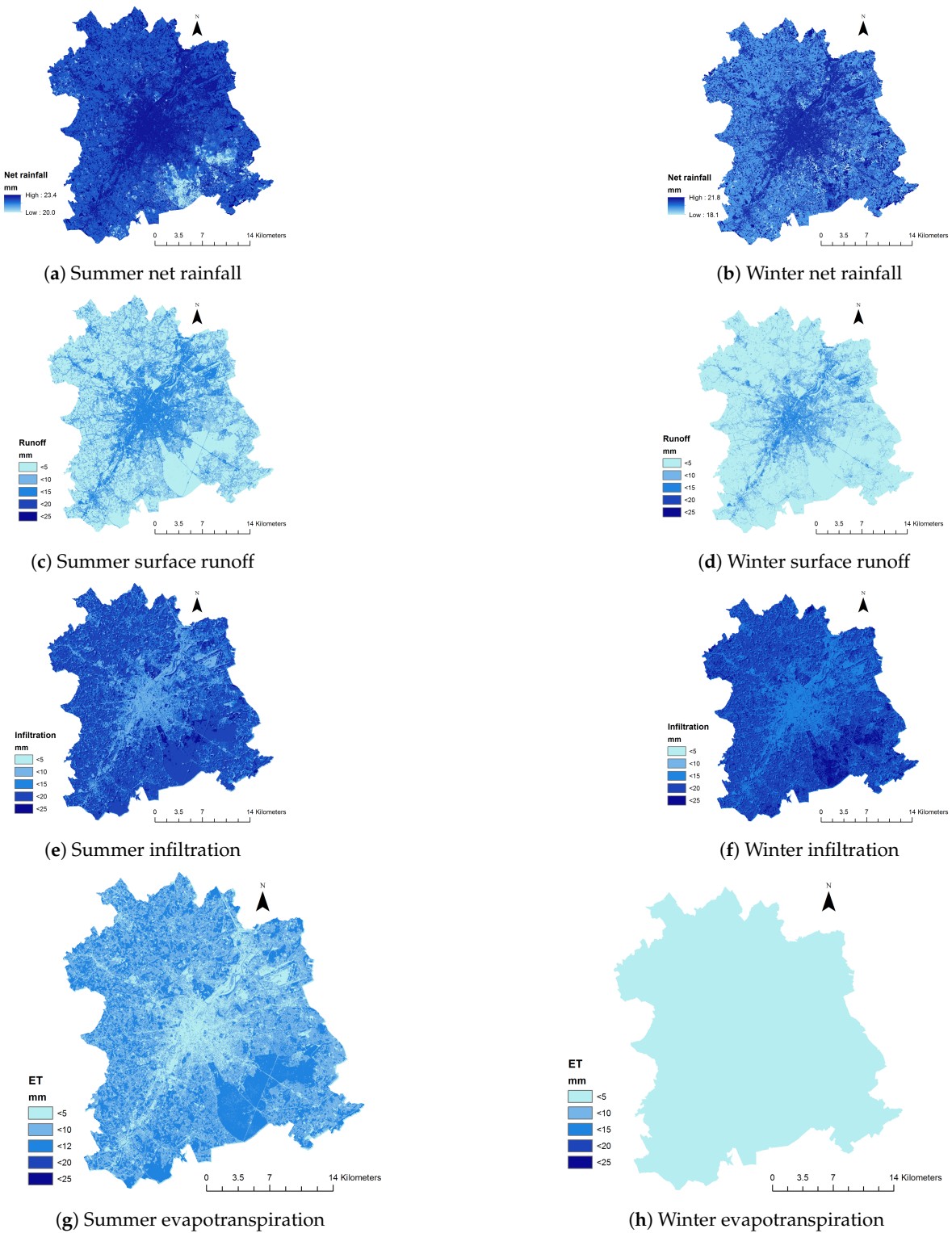

**Figure 16.** Water balance (20 m resolution) for an event in summer (24 mm/8 h) and winter (23 mm/41 h) in 2015.

## 4. Conclusions

Within this paper we quantified the hydrological response (net precipitation, runoff, infiltration and evapotranspiration) of a city using spatial and temporal metrics retrieved from remote sensing data.

Based on the remote sensing images, the heterogeneity of the urban surface has been characterized using 3 spatial metrics: the total impervious area, the vegetation cover and leaf area index to estimate (1) the net rainfall volume, (2) the runoff volume, (3) the infiltration volume and (4) the cumulative evapotranspiration. The routing of the surface runoff to the outlet (5) uses additional metrics based on the DEM. The best suited metric for this purpose is the timing to the outlet, assessed by means of the total impervious area and the hydrological distance to the outlet.

Compared to WetSpa simulations the remote sensing based metrics approach reaches $R^2$ values of 98% for net rainfall, 95% for surface runoff, 99% for infiltration and 97% for cumulative evapotranspiration. The routing in the Watermaelbeek catchment is evaluated with discharge observations and reaches an NSE value of 0.89 at a 2 m resolution. Further the hydrological response estimated at a 2 m resolution and validated with high resolution simulations and discharge data can be upscaled using 20 m resolution Sentinel satellite data ($R^2$ = 99%, NSE = 0.88). At a 20 m spatial resolution additional validation at a 10 min timestep reaches an NSE of 0.76 for the Watermalebeek catchment. Within the Roodebeek catchment the NSE equals to 0.73 at 20 m resolution and an hourly timestep.

To summarize, the proposed simple regression method is achieving comparable results to a distributed process based water balance model, for example, WetSpa. Similar to physically based models, the method preserves a correct mass balance and can be used for the evaluation of a single event as well as on a seasonal or even yearly time span using a timestep approach. On the other side, for our approach no model parameterization is needed to estimate the urban water balance and the computational time is close to 0. Furthermore, the input for our method is based on 'up-to-date' and open source remote sensing data which account for the heterogeneity of the urban surface. The production of detailed and distributed urban water balance maps at a city-wide scale facilitates visual interpretation and allows to identify landcover related differences in hydrological fluxes.

**Author Contributions:** Conceptualization, C.W. and B.V.; Funding acquisition, B.V.; Investigation, C.W.; Methodology, C.W.; Project administration, B.V.; Supervision, W.B. and B.V.; Validation, C.W.; Writing—original draft, C.W.; Writing—review & editing, W.B. and B.V.

**Funding:** This research is co-funded within the framework of the UrbanEARS project SR/00/307 from the Belgian Federal Science Policy Office, Support to the Exploitation and Research in Earth Observation III (BELSPO STEREOIII) and the Belgian airborne calibration and validation sites for urban and forest (BELAIR-SONIA) project SR/03/333. The meteorological dataset is provided by the Royal Meteorological Institute (Uccle) and Flowbru (depot communal, Roodebeek). The discharge dataset is provided by Flowbru (Watermaelbeek, Roodebeek).

**Acknowledgments:** We want to thank Frederik Priem for his contribution with regard to the landcover classifications of the APEX and Sentinel 2 images. We further acknowledge the constructive feedback of the 2 reviewers and the efficient editing.

**Conflicts of Interest:** The authors declare no conflict of interest.

## Abbreviations

The following abbreviations are used in this manuscript:

| | |
|---|---|
| APEX | Airborne Prism EXperiment |
| DCIA | Directly connected impervious area |
| Infil | Infiltration |
| LAI | Leaf area index |
| LIDAR | Light detection and ranging |
| NSE | Nash-Sutcliffe efficiency |
| Pbias | Percent bias |
| RDB | Roodebeek |
| RMSE | Root mean squarred error |
| RO | Surface runoff |
| RS | Remote sensing |

TIA      Total Impervious Area
Veg%     Vegetation cover
WetSpa   Water and Energy Transfer between Soil, Plants and Atmosphere
WMB      Watermaelbeek

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
