# Peer review of "Using Remote Sensing Based Metrics to Quantify the Hydrological Response in a City"

_water, doi:10.3390/w11091763_

Round 1

Reviewer 1 Report

The novelty of this article is solid, I believe the readership will be interested and that it is a good contribution to science. The manuscript does need a good amount of "polish" to be published. I think the approach is solid but needs more explanation for the reader to follow. Here are some overarching concerns:

There are many places where figure axis titles are confusing, where figure numbers are not in the right order, etc. The format of the figures could be more uniform as well. These are little things that would make the manuscript appear like more of a finished product

The methods are fairly sparse compared to the complexity shown in the results. In particular, clearing up the confusion on page 5 is important. The approach to only use one remote sensing variable in the regression equations, the inclusion of Pnet in each equation, etc., these are decisions by the authors that can be better explained and documented

The authors should push the novelty of the work in the intro and conclusions. Why do we need this approach? What are the benefits and shortcomings compared to other approaches like WetSpa? 

I have included a number of comments in the attached pdf for your consideration. 

Author Response

Dear reviewer, 

we thank you very much for your constructive feedback. We hereby answer to your comments using the line and figure numbers of the revised manuscript with track changes. Additionally we replied to the commented pdf in the attached file.

Best regards, the authors.

Reviewer comment

Authors reply

There are many places where figure axis titles are   confusing, where figure numbers are not in the right order, etc. The format   of the figures could be more uniform as well. These are little things that   would make the manuscript appear like more of a finished product

We did adjust the order of figures and tables. 

Sorry for the confusion.

Further we changed some figures:

Figure 4: The X-axis is called Pnet WetSpa

font sizes were adjusted

Figure 6: Legend adjusted

Figure 7: a) layout changed; b) y-axis title changed

Figure 8: x-axis was changed to be consistent with figures 11, 12, 14   and 15.

Figure 10: style adjusted to overall figure style of the paper/   quality increased

Figure 13: style of sub-plots adjusted to be more uniform

Figure 14: y-axis corrected and style adjusted to other discharge figures

The quality of figures 10, 11, 12, 15   was increased.

The methods are fairly sparse compared to the complexity shown in the   results. In particular, clearing up the confusion on page 5 is important. The   approach to only use one remote sensing variable in the regression equations,   the inclusion of Pnet in each equation, etc., these are decisions by the   authors that can be better explained and documented.

Part of the confusion occurred as tables and figures were not   positioned in the right order. We tried to reduce the confusion by adding   some more explanation in the results part: Ln 162-165

The authors should push the novelty of the work in the intro and   conclusions. Why do we need this approach? What are the benefits and   shortcomings compared to other approaches like WetSpa?

At the end of the results section (lines 320-350) we undergo a   discussion of the benefits and shortcomings of the method. In this paragraph   our approach is also compared to data-driven and process- or physically based   models such as WetSpa. To summarize, the remote sensing based metrics approach used here is a computationally very efficient approach to quantify the urban water balance with a physical meaning. Additionally the method isn't limited to the catchment borders but can be applied to communal borders. We therefore believe it is a useful tool for urban planning and   policy making with regards to sustainable urban (storm-)water management.

To stress the novelty in the paper a paragraph was added to the conclusion section (lines 369 - 377).

I have included a number of comments in the attached pdf for your consideration.

Thank you very much. 

We responded to the comments in the attached pdf.

Reviewer 2 Report

The paper uses remote sensing based metrics to quantify the hydrological response in a city.

The research uses data-driven approach to assess the efficiency of remote sensing data for accurately simulating parameters of a water balance model. 

 Very less information have been provided on the measured precipitation data- source of data, the quality control it has gone through. 

Also, authors need to justify why the WetSpa simulations are used for comparison?

Author Response

Dear reviewer,

we thank you very much for your feedback. Please see the attachment for our answers to your comments.

Best regards,

the authors

Round 2

Reviewer 1 Report

The authors have made many improvements to the text, including clarifying a number of concerns, reformatting figures/tables, etc. I suggest acceptance based on their responses.